# POST-HOC FEATURE SELECTION LAYER FOR NEURAL NETWORKS INTERPRETABILITY

## ABSTRACT

The interpretability of complex neural networks remains a critical challenge, especially for models already deployed in high-stakes domains. To address this, we introduce a post-hoc adaptation of the Feature Selection Layer (FSL). Our approach reframes the FSL as a lightweight, trainable module that integrates with already frozen pre-trained models on tabular datasets to highlight the features the original model considers most important. This post-hoc FSL learns relevance weights for input features by fine-tuning its weights based on the original model's learned outputs. Crucially, this process is non-invasive, operating without altering the original model's architecture or its learned parameters. We conducted our experiments using both statistical and visual metrics, including accuracy, F1 score, recall, precision, weighted t-SNE and silhouette score, and also analyzed the stability of the post-hoc FSL on high-dimensional synthetic and real-world tabular datasets. We compare the post-hoc FSL feature weighting method using these metrics against the original embedded FSL and other post-hoc interpretability methods, such as Integrated Gradients, Noise Tunnel, DeepLIFT, Gradient SHAP, and Feature Ablation. Experimental results demonstrate that post-hoc FSL feature weighting method successfully identified relevant features across the different datasets, maintaining the predictive power of the original neural network while enhancing its interpretability. While post-hoc FSL achieves similar predictive, visual and stability results comparable to the original FSL, it demonstrated distinct advantages over other state-of-the-art methods. Despite a trade-off in the Jaccard, Spearman and Pearson stability metrics, post-hoc FSL approach yielded, on average, superior performance on visual and clustering-based interpretability for real-world datasets, as measured by weighted t-SNE and the silhouette score.

## 1 INTRODUCTION

Over the past decade, the availability of massive amounts of data has positioned Deep Neural Networks (DNNs) as one of the most effective approaches for large-scale knowledge extraction. These models have achieved state-of-the-art performance in diverse domains such as computer vision, natural language processing, and healthcare by accurately diagnosing diseases and helping to anticipating them (Wason, 2018; Badawy et al., 2023; Khan et al., 2023). However, most DNNs operate as "black boxes," where the learned representations and decision-making processes remain opaque. The lack of interpretability presents serious challenges in high-stakes domains, especially in healthcare, where understanding a model's behavior is as crucial as its predictive accuracy. In contexts where medical diagnosis decisions and treatment recommendations are influenced by machine learning models, transparency becomes essential, not only for medical practitioners, but also for regulatory approval and ethical accountability (Miotto et al., 2018; Murad et al., 2024). Moreover, non-transparent models increase the risk of hidden biases, where illegitimate factors may influence predictions without detection. For instance, models trained on biased datasets may inadvertently exploit sensitive attributes to improve accuracy, ultimately leading to unfair or unsafe outcomes (Prince, 2023). Addressing the trade-off between accuracy and interpretability remains a critical challenge in the safe and reliable deployment of deep learning systems. To address this interpretability problem, feature weighting methods are a popular approach, assigning scores to input features based on their relevance to a model's predictions (Molnar et al., 2020). These methods are particularly valuable as they provide insight into feature behavior within complex models like DNNs,

especially in high-dimensional data problems. Among these techniques, the Feature Selection Layer (FSL), proposed by Figueroa Barraza et al. (2021), is an embedded method designed to learn feature relevance based on the model's internal dynamics during the training phase. This is achieved by attaching a new, trainable layer between the input and the main network. However, its embedded nature requires joint training from scratch, limiting its use for interpreting pre-trained DNN models. To overcome this limitation, we propose a post-hoc variant that preserves FSL's effectiveness while enabling feature relevance analysis on existing neural networks without altering their internal parameters, thereby enhancing interpretability and potentially improving predictive performance.

## 2 RELATED WORK

### 2.1 FEATURE SELECTION

Feature selection is a critical process for identifying an optimal subset of features to enhance model performance and reduce complexity (Barbieri et al., 2024). These techniques often rely on feature weighting to assign a relevance score to each feature (Tahir et al., 2007) and are typically categorized into several groups based on their interaction with the model training process (Miao & Niu, 2016). Filter methods are model-agnostic, ranking features via statistical tests as a preprocessing step. While computationally efficient, they often overlook feature interactions (Lazar et al., 2012). Wrapper methods leverage a specific predictive model to score and select feature subsets. This approach can yield high performance but is computationally expensive and tends to generalize poorly to different models (Barbieri et al., 2024). Embedded methods integrate feature selection directly into the model's training procedure. Similar to wrappers, the resulting feature set is model-specific and may not be optimal for other architectures (Barbieri et al., 2024). Hybrid approaches combine different feature selection methods to balance performance and efficiency (Ang et al., 2015), and Ensemble methods aggregate the results of multiple selection runs to produce a more robust and stable final feature set.

### 2.2 FEATURE SELECTION LAYER

The Feature Selection Layer (FSL), introduced by Figueroa Barraza et al. (2021), is an embedded feature weighting method for neural networks. It consists of a dense layer between the neural network's input and its first hidden layer, where each neuron maintains a one-to-one correspondence with an input feature. Weights, initialized as $1/n$, being $n$ the total number of features, are jointly trained with the network and reflect feature relevance during learning. A modified L1 regularization term is added to ensure the interaction between features, which is calculated as:

$$r(W^{\text{FSL}}) = \lambda \cdot \left| \sum_{t=1}^{n} W_t^{\text{FSL}} - 1 \right| \tag{1}$$

where $W^{\text{FSL}}$ represents the weights from the FSL, $n$ is the number of features and $\lambda$ is a value between 0 and 1 that determines the strength of the regularization. FSL them adds two hyperparameters to the network: the FSL activation function and the $\lambda$ term for regularization.

### 2.3 POST-HOC METHODS

Post-hoc feature attribution methods are pivotal for interpreting "black box" models such as neural networks. Prominent gradient techniques including Integrated Gradients (Sundararajan et al., 2017), DeepLIFT (Shrikumar et al., 2017), and Gradient SHAP (Lundberg & Lee, 2017) compute feature relevance relative to a reference input, or baseline. Integrated Gradients calculates relevance scores by integrating gradients along a linear path from the baseline to the input features. Similarly, Gradient SHAP utilizes gradients along such paths, but uses approximation to Shapley values to determine feature relevance. In contrast to these path-based approaches, DeepLIFT compares the activation of each neuron to its reference activation and backpropagates contribution scores based on this difference. Other prominent post-hoc interpretability methods include Feature Ablation (Kokhlikyan et al., 2020), a perturbation-based approach that evaluates feature importance based on the change in the model's output upon their ablation, and Noise Tunnel (Kokhlikyan et al., 2020), a technique designed to smooth feature attributions and reduce noise, thereby improving the reliability of gradient-based methods like Integrated Gradients. For our experiments we apply Noise Tunnel in conjunction

with Integrated Gradients. In this work, we evaluate the proposed post-hoc FSL variation of the FSL against these post-hoc methods using the metrics detailed in Section 4.2.

# 3 PROPOSED IMPLEMENTATION

We propose a novel post-hoc feature weighting method designed to enhance the interpretability of pre-trained neural networks. Our method adapts the Feature Selection Layer (FSL) from Figueroa Barraza et al. (2021), reformulating it as a trainable module applicable to any compatible pre-trained neural network. The core objective is to identify the most relevant features for a model's predictions without altering its original learned parameters.

## 3.1 ARCHITECTURE AND MECHANISM

Architecturally, the proposed post-hoc FSL remains faithful to the original FSL. It is a dense, trainable layer with a one-to-one mapping between neurons and input features, placed between the input and a pre-trained network (Figure 1). Each feature $\vec{x_i}$ is scaled by a trainable weight $\vec{w_i}$, producing a weighted vector $\vec{w}\vec{x}$ passed to the subsequent layers of the network. Despite its distinct training paradigm, we retain the inherent benefits of the FSL architecture, which is computationally efficient and performs competitively in dimensionality reduction and accuracy when compared to AFS (Gui et al., 2019), Random Forest, ReliefF, and Mutual Information (Figueroa Barraza et al., 2021).

## 3.2 POST-HOC TRAINING PROCEDURE

Our feature weighing approach operates via a distinct post-hoc training procedure. Given a pre-trained model $f_{\text{pre}}$ with its internal parameters frozen, a vector $\vec{x}$ that represents a single input, a vector $\vec{w}$ which represents the weights from the FSL, an activation function $a$, a loss function $\ell$, and a learning rate $\eta$, the training process is as follows:

**Step 1:** Our feature weighting layer is integrated between the input layer and the hidden layers from the pre-trained neural network (Equation 2).

**Step 2:** Data is fed through the combined model (FSL and the frozen network), and the final output is used to compute the loss function of the whole model (Equation 3).

**Step 3:** During backpropagation the parameters of the pre-trained network remain unchanged, gradients are computed only for the FSL weights.

**Step 4:** Steps 2 and 3 are repeated over several epochs. This allows the weights of the post-hoc FSL to converge, with their final magnitudes representing the feature importance.

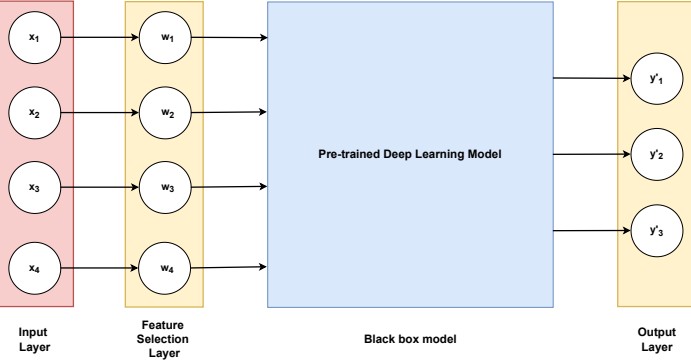

Figure 1: **Representation of post-hoc Feature Selection Layer.** In this example, four input features are individually multiplied by their activated weights within the FSL before being passed to the pre-trained model. During fine-tuning, only the weights within the FSL are updated, while all parameters of the pre-trained model remain frozen.

$$\vec{x}' = a(\vec{w}^{(i)}) \odot \vec{x} \qquad (2)$$

$$\vec{y} = f_{\text{pre}}(\vec{x}') \qquad (3)$$

### 3.3 Layer Configuration: Regularization, Activation, and Initialization

We adopt three design choices to regulate the Feature Selection Layer (FSL). First, we apply the standard $L1$ regularization to promote sparsity by shrinking weights of irrelevant features toward zero. Second, we use ReLU activation to ensure non-negative weights, simplifying interpretation, a weight of zero indicates irrelevance, and positive values represent feature's importance. Third, we initialize weights to $1.0$ instead of $1/n$, providing a stable starting point in which all features are ingested by the pre-trained model initially using unchanged values and preventing initializations with weights close to zero. This allows the optimization process to more effectively distinguish between features to retain and those to discard, ultimately improving both training dynamics and model performance.

## 4 Experiments

This section details the experimental setup used to evaluate the proposed post-hoc FSL method. We describe the datasets (Section 4.1), the evaluation metrics (Section 4.2), and the specific implementation details (Section 4.3).

### 4.1 Datasets

To evaluate the proposed feature selection layer, we employed a diverse range of datasets to analyze its performance across various relevant challenges, such as high dimensionality, low sample sizes, and the presence of noisy features. To this end, we utilized both synthetic (Section 4.1.1) and real-world datasets (Section 4.1.2). The synthetic data is composed of a pre-defined number of relevant and noisy features. The real-world datasets include an email spam classification dataset (Balaka, 2020) and gene expression microarray data (Feltes et al., 2019), which are characteristically high-dimensional with a low number of samples. The purpose of using this variety of datasets is to comprehensively assess post-hoc FSL's ability to reduce dimensionality and, consequently, enhance the interpretability of neural network models by highlighting salient features while filtering out noisy ones. All datasets are fully described in Table 7 at Appendix A.

### 4.1.1 Synthetic Datasets

For this analysis, we used two synthetic datasets with controlled proportions of relevant and irrelevant features. The XOR dataset (Barbieri et al., 2024) contains 500 samples and 50 features, with two informative and 48 noisy ones. SynthA includes $3,000$ samples and $100$ features, of which $30$ are relevant. It was generated by first defining an $n$-dimensional hypercube, where $n$ is the number of informative features. Then, Gaussian clusters of data points are created at each vertex of the hypercube, with an equal number of clusters assigned to each class. The informative features are the ones that are coordinates for the created points in the hypercube (Barbieri et al., 2024). Synthetic data provides ground truth for feature relevance, enabling direct evaluation of post-hoc FSL's selection accuracy (Section 4.2), based on the weights assigned to each feature.

### 4.1.2 Real-world datasets

To evaluate the post-hoc FSL on real-world high dimensionality and low sample sizes (HDLSS) challenges, we selected datasets from two domains. First, we utilized microarray datasets from the CuMiDa repository (Feltes et al., 2019), a database of Homo sapiens datasets selected from over $30,000$ GEO experiments, following bias-reduction guidelines by Grisci et al. (2024), we used Liver-GSE22405 (48 samples, $22,284$ features, binary classification) and Breast-GSE45827 (151 samples, $54,676$ features, six-class classification). Additionally, we evaluated a spam email dataset (Balaka, 2020) with $5,172$ samples and $3,000$ features, representing word frequencies in the email corpus.

## 4.2 EVALUATION METRICS

To evaluate the post-hoc FSL method, we employed several metrics to assess its effectiveness. We group these into three categories: statistical metrics (Section 4.2.1), visual analysis (Section 4.2.2), and stability (Section 4.2.3).

### 4.2.1 PERFORMANCE METRICS

We assess model predictive performance using accuracy, precision, recall, and F1 score. To evaluate feature selection, we leverage synthetic datasets with known ground truth and apply two metrics (Barbieri et al., 2024): **Percentage of Informative Features Selected (PIFS)** measures the proportion of informative features captured in a selected subset (Equation 4); **Percentage of Selected Features that are Informative (PSFI)** quantifies how many selected features are truly informative (Equation 5). Each metric spans from $0.0$, indicating no informative features were selected, to $1.0$, reflecting ideal selection.

$$PIFS = \frac{|S_{\text{selected}} \cap S_{\text{informative}}|}{|S_{\text{informative}}|} \tag{4}$$

$$PSFI = \frac{|S_{\text{selected}} \cap S_{\text{informative}}|}{|S_{\text{selected}}|} \tag{5}$$

### 4.2.2 VISUAL METRICS

For visual analysis, we utilize a variation of the t-SNE (t-Distributed Stochastic Neighbor Embedding), a visualization algorithm originally proposed by (Maaten & Hinton, 2008). t-SNE projects high-dimensional data into a low-dimensional space (typically two dimensions) by calculating the pairwise distances between data points, commonly using the Euclidean distance. To specifically visualize the discriminative power of the features identified by the post-hoc FSL, we employ a variant known as weighted t-SNE (Grisci et al., 2025). This approach introduces a weighted Euclidean distance to enhance the visualization based on the feature weights provided by the feature weighting algorithm, which range from zero (less relevant) to one (more relevant). When scaling each dimension by its weight, dimensions with higher weights contribute more significantly to the distance between points and thus have a greater influence on their final positions. To assess how feature weights affect cluster separability, we used the silhouette coefficient (Rousseeuw, 1987), which ranges from $-1$ to $1$. Higher values indicate well-separated and dense clusters, while values near $0$ suggest boundary points and values close to $-1$ indicates that the points are closer to some other cluster than the one they are assigned.

### 4.2.3 STABILITY METRICS

To evaluate the stability of the post-hoc FSL feature weighting method, we incorporated a stability analysis protocol. The goal is to evaluate how the method's output changes with small perturbations in the training data. Data perturbation is defined as changes in the training samples and can be simulated by adding, removing, or re-sampling instances (Awada et al., 2012). For this purpose, we employed $k$-fold Cross-Validation, a widely used technique to evaluate model performance and measure the stability of feature selection methods (Awada et al., 2012). The implementation of this protocol begins by partitioning the dataset into $k$ stratified folds. The process then iterates $k$ times; in each iteration, the feature weighting method is applied to a training set composed of $k$-1 folds, while a different fold is held out. The feature weights generated in each iteration are stored. After completing all $k$ iterations, this procedure yields $k$ distinct sets of feature weights, allowing us to evaluate the stability of the feature weighting method by analyzing the consistency across these results. To assess the consistency of the feature weighting algorithms, we apply three metrics across the $k$ sets of feature weights proposed by Barbieri et al. (2024). The Jaccard Index measures the overlap between two sets with the top-$n$ selected features, capturing the subset stability. The Person correlation evaluates the similarity of feature weights. Lastly, the Spearman rank correlation quantifies the agreement between feature rankings. All metric equations are detailed in Appendix B.

### 4.3 EXPERIMENTS SETUP

For our experimental evaluation, we trained two distinct models: (1) a baseline model without any embedded feature weighting mechanism and (2) a model with an embedded Feature Selection Layer (FSL). The baseline model was designed to be interpreted by the proposed post-hoc FSL and other post-hoc methods, as described in Section 2.3. To assess post-hoc FSL effectiveness, we compared it against state-of-the-art post-hoc methods introduced in Section 2.3 and the original FSL method using the statistical, visual, and stability metrics from Section 4.2.

## 5 RESULTS AND DISCUSSION

### 5.1 RESULTS FOR SYNTHETIC DATASET

When comparing the predictive performance for the XOR dataset, the baseline model achieved near-optimal predictive performance, 0.999 across all metrics, while both FSL and post-hoc FSL reached perfect scores of 1.0. All models consistently identified the two relevant features across folds, as shown by PIFS, PSFI, Jaccard, and Pearson metrics (Table 1 and Table 8 of Appendix C). Despite this, low Spearman correlations indicate that while the algorithms successfully prioritized relevant features over irrelevant ones, the relative positions among informative features, as well as among irrelevant ones, can vary. Silhouette scores (Table 2) and weighted t-SNE visualizations (Figure 4, Appendix C) confirmed improved class separability when compared to the standard t-SNE, though no method showed clear superiority.

Table 1: **Results for stability metrics using the XOR dataset.** Bold indicates the highest average. Jaccard was calculated using the top-2 weighted features.

| Model | Jaccard | Spearman | Pearson |
|---|---|---|---|
| Integrated Gradients | $1.000 \pm 0.000$ | $0.151 \pm 0.114$ | $0.996 \pm 0.001$ |
| Noise Tunnel | $1.000 \pm 0.000$ | $\mathbf{0.162 \pm 0.111}$ | $0.987 \pm 0.004$ |
| Deep Lift | $1.000 \pm 0.000$ | $0.153 \pm 0.114$ | $0.996 \pm 0.001$ |
| Gradient SHAP | $1.000 \pm 0.000$ | $0.158 \pm 0.112$ | $0.990 \pm 0.003$ |
| Feature Ablation | $1.000 \pm 0.000$ | $0.159 \pm 0.107$ | $\mathbf{0.998 \pm 0.001}$ |
| FSL | $1.000 \pm 0.000$ | $0.111 \pm 0.132$ | $0.988 \pm 0.007$ |
| Post-hoc FSL | $1.000 \pm 0.000$ | $0.126 \pm 0.140$ | $0.988 \pm 0.010$ |

Table 2: **Silhouette scores using the XOR and SynthA datasets.** Bold indicates the highest average; best statistically significant results ($p < 0.01$) from Kruskal-Wallis and Dunn's tests are marked in orange.

| Model | XOR Silhouette Score | SynthA Silhouette Score |
|---|---|---|
| None | $0.003 \pm 0.002$ | $0.043 \pm 0.000$ |
| Integrated Gradients | $0.203 \pm 0.002$ | $\mathbf{0.235 \pm 0.049}$ |
| Noise Tunnel | $0.201 \pm 0.002$ | $0.177 \pm 0.083$ |
| Deep Lift | $0.203 \pm 0.002$ | $0.233 \pm 0.064$ |
| Gradient SHAP | $0.203 \pm 0.002$ | $0.194 \pm 0.083$ |
| Feature Ablation | $\mathbf{0.205 \pm 0.002}$ | $0.181 \pm 0.076$ |
| FSL | $\mathbf{0.205 \pm 0.001}$ | $0.191 \pm 0.090$ |
| Post-hoc FSL | $0.204 \pm 0.003$ | $0.115 \pm 0.077$ |

Table 3 reveals a performance disparity between FSL and post-hoc FSL on the SynthA dataset. Although post-hoc FSL moderately improved over the baseline, FSL consistently achieved higher predictive metrics. Moreover, the post-hoc FSL method underperformed both the FSL and other post-hoc approaches, as evidenced by the weighted t-SNE visualization presented in Figure 5 of Appendix C and the silhouette scores in Table 2, yielding lower average values and reduced stability

metrics. Despite these limitations, it reached PFSI and PIFS scores of $0.906$ on the top-30 features (Table 8 of Appendix C), demonstrating its ability to identify informative features, albeit, in this specific case, less effectively than the FSL, which reached perfect scores of $1.0$ on both metrics. These results also influenced post-hoc FSL stability scores (Table 4), which also had an inferior performance when compared to the other methods. We hypothesize that the synthetic dataset's structure may have hindered post-hoc FSL's feature identification. Nonetheless, its performance on real-world, high-dimensional datasets remained strong.

Table 3: **Prediction performance using the SynthA dataset.** Bold indicates the highest average; best statistically significant results ($p < 0.01$) from Kruskal-Wallis and Dunn's tests are marked in orange.

| Model | F1 Score | Accuracy | Precision | Recall |
|---|---|---|---|---|
| Baseline | $0.890 \pm 0.010$ | $0.890 \pm 0.009$ | $0.892 \pm 0.009$ | $0.890 \pm 0.009$ |
| FSL | $\mathbf{0.954 \pm 0.007}$ | $\mathbf{0.954 \pm 0.007}$ | $\mathbf{0.954 \pm 0.006}$ | $\mathbf{0.954 \pm 0.007}$ |
| FSL post-hoc | $0.916 \pm 0.009$ | $0.916 \pm 0.009$ | $0.917 \pm 0.009$ | $0.916 \pm 0.009$ |

Table 4: **Results for stability metrics using the SynthA dataset.** Bold indicates the highest average. Jaccard was calculated using the top-30 weighted features.

| Model | Jaccard | Spearman | Pearson |
|---|---|---|---|
| Integrated Gradients | $0.855 \pm 0.028$ | $0.758 \pm 0.038$ | $0.969 \pm 0.009$ |
| Noise Tunnel | $0.954 \pm 0.031$ | $0.752 \pm 0.038$ | $0.974 \pm 0.007$ |
| Deep Lift | $0.976 \pm 0.034$ | $0.760 \pm 0.037$ | $0.971 \pm 0.008$ |
| Gradient SHAP | $0.939 \pm 0.041$ | $0.750 \pm 0.041$ | $0.975 \pm 0.006$ |
| Feature Ablation | $0.956 \pm 0.050$ | $0.759 \pm 0.037$ | $0.980 \pm 0.005$ |
| FSL | $\mathbf{1.000 \pm 0.000}$ | $\mathbf{0.805 \pm 0.030}$ | $\mathbf{0.980 \pm 0.003}$ |
| Post-hoc FSL | $0.759 \pm 0.052$ | $0.759 \pm 0.030$ | $0.770 \pm 0.027$ |

To complement our experiments, we incorporated TabPFN, a pre-trained neural network model introduced by Hollmann et al. (2022), to evaluate the performance of post-hoc FSL. TabPFN is a Transformer-based architecture designed for supervised classification on small tabular datasets. It operates efficiently without requiring hyperparameter tuning and demonstrates competitive performance against state-of-the-art classification methods. Given the model's complexity and time constraints, we restricted this experiment to the SynthA dataset. Additionally, due to source code compatibility, comparisons were made using the original Kernel SHAP implementation. Both post-hoc techniques successfully identified the most relevant features, yielding identical PSFI and PIFS scores of $0.966$ when evaluating the top-30 ranked features, which ideally should include all, and only, the informative features. When comparing the generated rankings, both methods assigned greater importance to the noisy feature "noisy_85" while undervaluing the informative feature "informative_24". Nonetheless, post-hoc FSL outperformed both weighted t-SNE using Kernel SHAP weights and standard t-SNE (Figure 2), achieving a higher silhouette score of $0.316$ compared to $0.186$ and $0.070$, respectively.

## 5.2 RESULTS FOR REAL-WORLD DATASETS

We evaluated the performance of the post-hoc FSL variation on real-world datasets, as described in Section 4.1.2. These datasets comprise tasks in spam email detection and microarray analysis, the latter being characterized by high dimensionality and a low sample size. The results for the Spam dataset are presented in Table 5, 6 and Figure 3. As shown in Table 5, the baseline and post-hoc FSL models achieved the highest performance metrics. The baseline model, in particular, recorded the best average scores across accuracy, F1 score, recall, and precision. While the baseline model demonstrated the top performance, post-hoc FSL obtained comparable results and outperformed FSL. In terms of visual separability metrics, the FSL post-hoc yielded the best results, achieving a

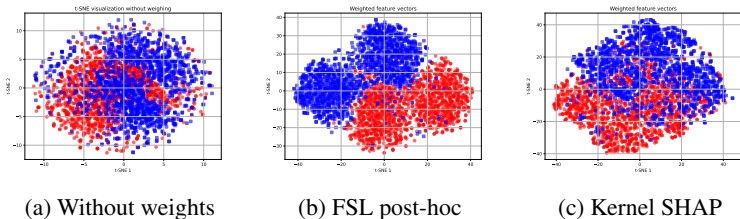

| (a) Without weights | (b) FSL post-hoc | (c) Kernel SHAP |

Figure 2: **Weighted t-SNE of the post-hoc FSL and Kernel SHAP methods using the TabPFN model** SynthA dataset was used for the experiment.

higher average silhouette coefficient than both the FSL and the other post-hoc methods. However, regarding stability, both, post-hoc FSL and FSL, were inferior to the other post-hoc approaches. Although the method exhibits limited stability across different runs, it consistently assigns meaningful weights to a subset of the most relevant features. This suggests that, while the overall feature ranking may vary, the features identified as important by the method tend to be informative. A more in-depth analysis of the stability metrics is provided in Appendix D.

Table 5: **Prediction performance for real-world datasets.** Bold indicates the highest average; best statistically significant results ($p < 0.01$) from Kruskal-Wallis and Dunn's tests are marked in orange.

(a) Spam dataset

| Model | F1 Score | Accuracy | Precision | Recall |
|---|---|---|---|---|
| Baseline | $\mathbf{0.976 \pm 0.005}$ | $\mathbf{0.976 \pm 0.005}$ | $\mathbf{0.977 \pm 0.005}$ | $\mathbf{0.976 \pm 0.005}$ |
| FSL | $0.968 \pm 0.004$ | $0.968 \pm 0.004$ | $0.969 \pm 0.004$ | $0.968 \pm 0.004$ |
| FSL post-hoc | $0.973 \pm 0.003$ | $0.973 \pm 0.003$ | $0.974 \pm 0.003$ | $0.973 \pm 0.003$ |

(b) Liver dataset

| Model | F1 Score | Accuracy | Precision | Recall |
|---|---|---|---|---|
| Baseline | $0.800 \pm 0.076$ | $0.800 \pm 0.076$ | $0.858 \pm 0.079$ | $0.800 \pm 0.076$ |
| FSL | $\mathbf{0.840 \pm 0.083}$ | $\mathbf{0.840 \pm 0.083}$ | $\mathbf{0.893 \pm 0.055}$ | $\mathbf{0.840 \pm 0.083}$ |
| Post-hoc FSL | $0.827 \pm 0.070$ | $0.827 \pm 0.070$ | $0.884 \pm 0.047$ | $0.827 \pm 0.070$ |

(c) Breast dataset

| Model | F1 Score | Accuracy | Precision | Recall |
|---|---|---|---|---|
| Baseline | $0.942 \pm 0.024$ | $0.945 \pm 0.021$ | $0.955 \pm 0.024$ | $0.945 \pm 0.021$ |
| FSL | $0.955 \pm 0.033$ | $0.958 \pm 0.030$ | $0.964 \pm 0.030$ | $0.958 \pm 0.030$ |
| FSL post-hoc | $\mathbf{0.970 \pm 0.033}$ | $\mathbf{0.970 \pm 0.032}$ | $\mathbf{0.978 \pm 0.024}$ | $\mathbf{0.970 \pm 0.032}$ |

Analysis of the Liver and Breast microarray datasets shows that post-hoc FSL achieved comparable results to the baseline and the FSL when compared to their predictive power (Table 5). However, the post-hoc FSL and the FSL methods exhibit significantly superior performance in generating meaningful feature attributions. When evaluating the quality of feature selection through weighted t-SNE visualizations (Figure 6, 7 of Appendix C) with their respective silhouette coefficients (Table 6), both FSL methods outperformed all other state-of-the-art post-hoc techniques. On the Liver dataset, the post-hoc FSL variant yielded the highest silhouette score. In contrast, the FSL was the top performer on the Breast dataset. These results underscore the FSL's capacity to identify highly relevant features that lead to better-defined class clusters.

Table 6: **Silhouette score for real-world dataset.** Bold indicates the highest average; best statistically significant results ($p < 0.01$) from Kruskal-Wallis and Dunn's tests are marked in orange.

| Model | Spam | Liver | Breast |
|---|---|---|---|
| None | $0.090 \pm 0.000$ | $0.066 \pm 0.000$ | $0.192 \pm 0.000$ |
| Integrated Gradients | $0.202 \pm 0.031$ | $0.170 \pm 0.031$ | $0.451 \pm 0.036$ |
| Noise Tunnel | $0.189 \pm 0.034$ | $0.178 \pm 0.044$ | $0.447 \pm 0.045$ |
| Deep Lift | $0.202 \pm 0.025$ | $0.172 \pm 0.049$ | $0.449 \pm 0.030$ |
| Gradient SHAP | $0.209 \pm 0.022$ | $0.199 \pm 0.042$ | $0.450 \pm 0.040$ |
| Feature Ablation | $0.196 \pm 0.029$ | $0.181 \pm 0.030$ | $0.450 \pm 0.033$ |
| FSL | $0.234 \pm 0.013$ | $0.574 \pm 0.141$ | $\mathbf{0.647 \pm 0.021}$ |
| Post-hoc FSL | $\mathbf{0.235 \pm 0.069}$ | $\mathbf{0.666 \pm 0.012}$ | $0.511 \pm 0.078$ |

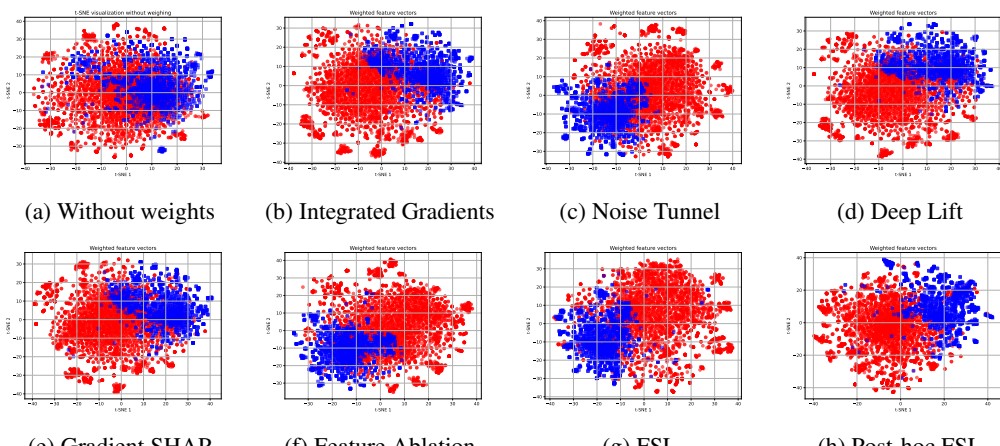

(a) Without weights  (b) Integrated Gradients  (c) Noise Tunnel  (d) Deep Lift

(e) Gradient SHAP  (f) Feature Ablation  (g) FSL  (h) Post-hoc FSL

Figure 3: **Weighted t-SNE** of all feature weighing metrics using the Spam dataset.

## 6 CONCLUSION

This paper explores the development of a variation of the Feature Selection Layer (FSL) as a post-hoc feature weighing method to improve the interpretability of pre-trained Deep Neural Networks. Our work demonstrates that the proposed method not only achieves high performance in identifying relevant features in high-dimensional datasets but also shows performance comparable to, and in several cases surpassing, other state-of-the-art post-hoc feature weighting methods.

### 6.1 LIMITATIONS AND FUTURE WORK

While our method proved to be an efficient feature weighing method when used in binary classification datasets, it showed unstable perfomance when dealing with multi-class classification problems, as it was unable to capture local and class-specific relationships with features. Futhermore, our proposed method is specifically designed for tabular data, where each feature has a fixed meaning within the dataset. However, this is not the case for images, where meaning of each pixel can vary across samples. Future research should therefore focus on adapting our post-hoc method to address these conditions, thereby extending its interpretability capabilities to such domains.

### ETHICS STATEMENT

The authors state they read the ICLR Code of Ethics and adhere to it.

## REPRODUCIBILITY STATEMENT

**Paper source code** is available at [The link will be shared after the reviewing process.] with all necessary dependencies and steps to reproduce each of the experiments presented in this paper. The original code of the weighted t-SNE (Grisci et al., 2025) can be found at https:github.com/sbcblab/weighted_tSNE. **CuMiDa** datasets (Feltes et al., 2019) can be found at https:sbcb.inf.ufrgs.br/cumida and **Spam** dataset (Balaka, 2020) can be found at https://www.kaggle.com/datasets/balaka18/email-spam-classification-dataset-csv. Both **XOR** and **SynthA** dataset are available at the same repository of the paper source code. **TabPFN source code** is available at https://github.com/PriorLabs/TabPFN and **Kernel SHAP documentation** is available at https://shap.readthedocs.io/en/latest/.

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

APPENDIX

## A    DATASETS DESCRIPTION

Table 7 describes each dataset used on the present work experiments and how many folds were created to conduce stability experiments. All selected synthetic and real-world datasets consist of complete numerical features with no missing values. The choice of the microarray datasets from CuMiDa repository is particularly advantageous as it not only provides datasets with the desired HDLSS characteristics but also offers baseline accuracy scores from other machine learning techniques, serving as a valuable point of comparison.

Table 7: **Datasets used for the experiments**

| **Name** | **Feature type** | **Labels** | **Features** | **Informative** | **Folds used** | **Samples** |
|---|---|---|---|---|---|---|
| XOR | Binary | 2 | 50 | 2 | 11 | 500 |
| SynthA | Numerical | 2 | 100 | 30 | 11 | 3000 |
| Spam | Numerical | 2 | 3000 | Unknown | 11 | 5172 |
| Liver-22405 | Numerical | 2 | 22284 | Unknown | 15 | 48 |
| Breast-45827 | Numerical | 6 | 54676 | Unknown | 15 | 151 |

## B    STABILITY METRICS

The Jaccard Index (Equation 6) measures the overlap between top-$n$ selected features from two sets $A$ and $B$, capturing subset stability. The Pearson correlation coefficient (Equation 7) evaluates the similarity of feature weights, where $x$ and $y$ are random variables, $x_i$ and $y_i$ are their respective $i$th values, $n$ is the number of observations, and $\bar{x}$, $\bar{y}$ are their means. Lastly, the Spearman rank correlation (Equation 8) quantifies the agreement between feature rankings, with $p$ and $q$ representing ranked permutations and $n$ the number of elements, thereby capturing the stability of rank order across selections.

$$J(A, B) = \frac{|A \cap B|}{|A \cup B|} \tag{6}$$

$$r_{xy} = \frac{\sum_{i=1}^{n}(x_i - \bar{x})(y_i - \bar{y})}{\sqrt{\sum_{i=1}^{n}(x_i - \bar{x})^2}\sqrt{\sum_{i=1}^{n}(y_i - \bar{y})^2}} \tag{7}$$

$$r_s(p, q) = 1 - \frac{6\sum_{i=1}^{n}(p_i - q_i)^2}{n(n^2 - 1)} \tag{8}$$

## C    COMPLEMENTARY RESULTS FROM EXPERIMENTS

Figure 4 presents the weighted t-SNE visualizations for the XOR dataset, complementing the silhouette scores reported in Table 2. All methods improved cluster separation relative to the unweighted baseline shown in Figure 4a, suggesting successful identification of the two informative features. This is confirmed in Table 8, where all algorithms achieved perfect PIFS and PSFI scores of 1.0 when selecting the top two features, indicating consistent detection of the relevant inputs. The same table also reports results for the SynthA dataset, where the top-30 features were used as this correspond to the known set of informative features. FSL was the only method to achieve a perfect score of 1.0, while post-hoc FSL reached 0.9, the lowest among the evaluated approaches. This discrepancy is reflected in the silhouette scores in Table 2, where post-hoc FSL also recorded the lowest value among feature weighting methods. Nonetheless, it successfully identified the majority of relevant features, contributing to improved cluster visualization in the weighted t-SNE visualization shown in Figure 5.

Figure 6 and Figure 7 complement the silhouette scores reported in Table 6 for the Liver and Breast datasets. In the Liver dataset, both FSL and post-hoc FSL improved cluster separation between the two classes, with post-hoc FSL achieving the highest silhouette score. The reason behind the highest silhouette score is visually discernible. For the Breast dataset, the visualization is more complex due to the larger number of classes. However, FSL produced the most distinct clustering, as reflected in its superior silhouette score. Post-hoc FSL ranked second, and despite the increased difficulty in visual interpretation, its performance remains visually distinguishable from the others.

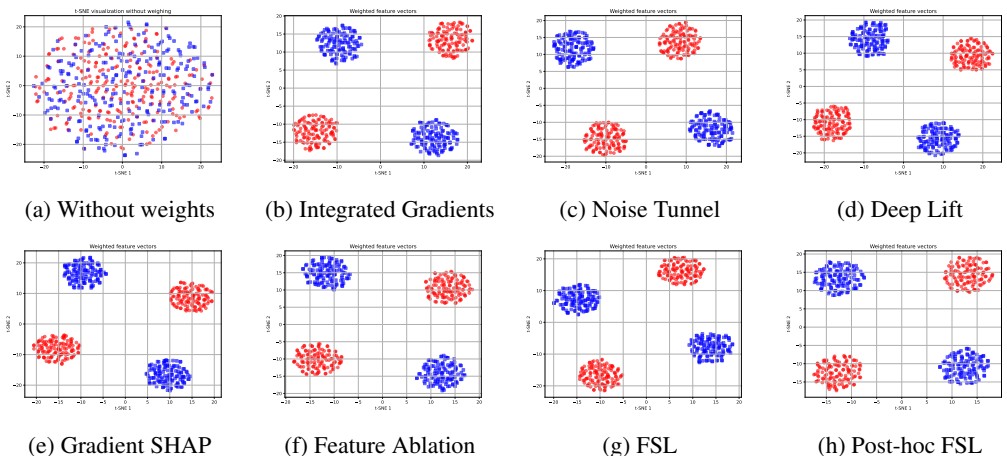

|        (a) Without weights        |        (b) Integrated Gradients        |        (c) Noise Tunnel        |        (d) Deep Lift        |
|        (e) Gradient SHAP          |        (f) Feature Ablation            |        (g) FSL                |        (h) Post-hoc FSL      |

Figure 4: **Weighted t-SNE** of all feature weighing metrics using the XOR dataset.

Table 8: **Comparison of PIFS and PFSI scores for the XOR and SynthA datasets.** The metrics were calculated using the subset of the top-2 most weighted features for the XOR dataset, while the SynthA dataset was calculate using the top-30.

|                       | XOR |      | SynthA |      |
|-----------------------|------|------|--------|------|
| Model                 | PIFS | PFSI | PIFS   | PFSI |
| Integrated Gradients  | 1.00 | 1.00 | 0.99   | 0.99 |
| Noise Tunnel          | 1.00 | 1.00 | 0.97   | 0.97 |
| Deep Lift             | 1.00 | 1.00 | 0.99   | 0.99 |
| Gradient SHAP         | 1.00 | 1.00 | 0.96   | 0.96 |
| Feature Ablation      | 1.00 | 1.00 | 0.99   | 0.99 |
| FSL                   | 1.00 | 1.00 | 1.00   | 1.00 |
| Post-hoc FSL          | 1.00 | 1.00 | 0.90   | 0.90 |

## D  STABILITY METRICS ANALYSIS

As discussed in Section 5.1, although all algorithms effectively prioritized relevant features over irrelevant ones, the relative ranking of features, both informative and non-informative, varied across executions. Unlike synthetic datasets, real-world datasets lack ground truth for feature relevance, preventing the use of Jaccard scores based on the exact number of informative features. Focusing on stability metrics for the Spam dataset (Table 9), the Jaccard scores were relatively consistent across methods, indicating stable selection of features within the top-100 ranking across multiple runs. In contrast, Spearman scores, which assess rank-order consistency, revealed greater instability for FSL and post-hoc FSL, suggesting that their feature rankings fluctuated more significantly between executions. Pearson scores were generally higher across all methods, likely due to the large number of consistently irrelevant features with near-zero weights. This behavior is illustrated in Figure 8, which tracks feature rank changes across executions. Relevant features tended to remain within the top-30 positions, while irrelevant ones clustered near the bottom. However, it is also possible to notice a considerable variation between these regions, particularly for post-hoc FSL, which exhibited the lowest stability scores, while FSL exhibited the highest stability scores (Table 4).

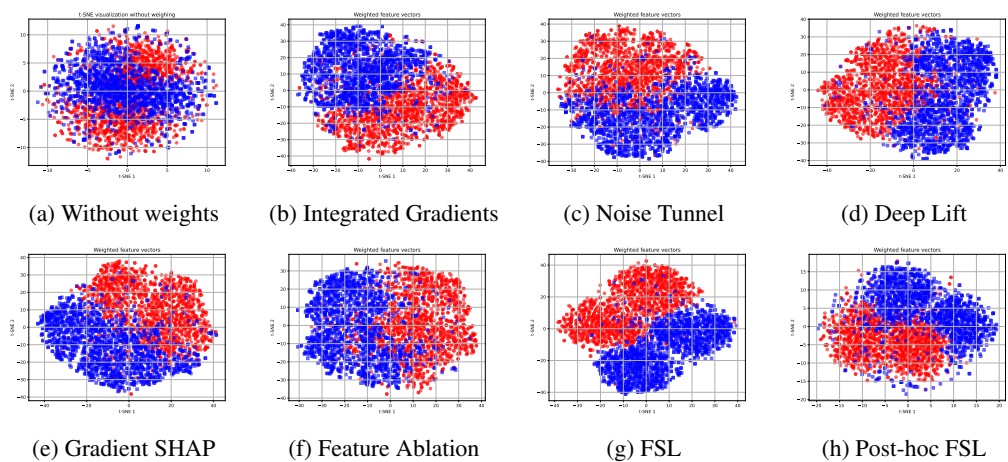

Figure 5: **Weighted t-SNE of all feature weighing metrics using the SynthA dataset.**

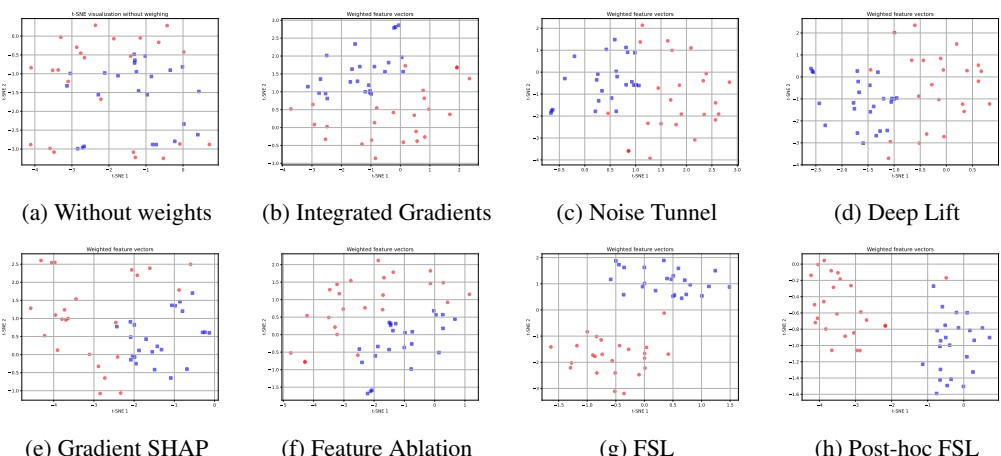

Figure 6: **Weighted t-SNE of all feature weighing metrics using the Liver dataset.**

Another important consideration for multi-class datasets, as Breast, is that FSL and post-hoc FSL computes a single, global importance weight for each feature based on its aggregated contribution across all classes. This global attribution approach is inherently less effective in complex multi-class settings, where a feature's relevance is often highly context-dependent-critical for predicting certain classes but not others. Most of the post-hoc methods, and all the ones that we used to compare with post-hoc FSL on this work, are capable to individually assess feature contributions for each specific class. This approach successfully captures local and class-specific relationships, which, when aggregated, produce a more robust and meaningful global attribution.

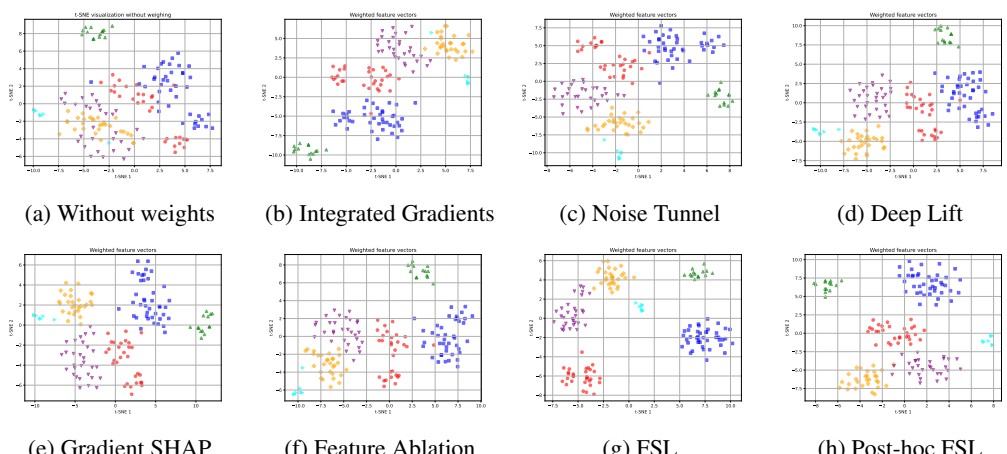

(a) Without weights    (b) Integrated Gradients    (c) Noise Tunnel    (d) Deep Lift

(e) Gradient SHAP    (f) Feature Ablation    (g) FSL    (h) Post-hoc FSL

Figure 7: **Weighted t-SNE of all feature weighing metrics using the breast dataset.**

Table 9: **Results for stability metrics using the spam dataset.** In bold is the highest average of each metric. Jaccard was calculated using the top-100 weighted features.

| Model | Jaccard | Spearman | Pearson |
|---|---|---|---|
| Integrated Gradients | $0.482 \pm 0.041$ | $0.739 \pm 0.028$ | $0.817 \pm 0.025$ |
| Noise Tunnel | $0.431 \pm 0.047$ | $0.553 \pm 0.036$ | $0.725 \pm 0.029$ |
| Deep Lift | $0.480 \pm 0.041$ | $0.738 \pm 0.028$ | $0.816 \pm 0.025$ |
| Gradient SHAP | $0.483 \pm 0.040$ | $0.698 \pm 0.028$ | $0.807 \pm 0.026$ |
| Feature Ablation | $\mathbf{0.486 \pm 0.043}$ | $\mathbf{0.741 \pm 0.028}$ | $\mathbf{0.818 \pm 0.025}$ |
| FSL | $0.397 \pm 0.029$ | $0.157 \pm 0.021$ | $0.728 \pm 0.022$ |
| Post-hoc FSL | $0.382 \pm 0.039$ | $0.312 \pm 0.024$ | $0.614 \pm 0.084$ |

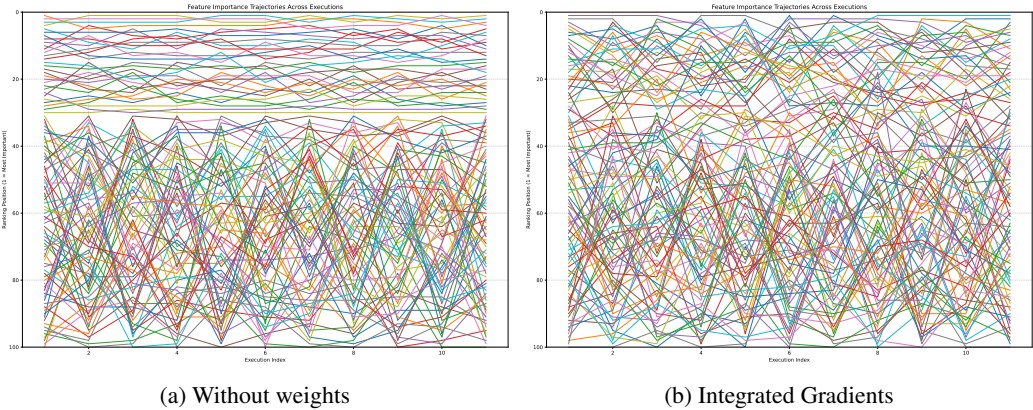

(a) Without weights      (b) Integrated Gradients

Figure 8: **Feature ranking between different executions on SynthA dataset.** All rankings were created from the feature weights in descending order. Each algorithm was executed 11 times. The vertical axis represents each individual execution, while the horizontal axis shows the ranking position of each feature, from 1 (top) to 300 (bottom), within that run. Different colors represent different features. A clear distinction emerged, with relevant features consistently ranking at the top positions, while noisy features fluctuate across positions at the bottom of the graph.

