# OpenReview forum: "Post-Hoc Feature Selection Layer for Neural Networks Interpretability"
_ICLR.cc/2026/Conference — Submitted to ICLR 2026_

### Official Review · Reviewer_SciF · 2025-10-31

**Soundness:** 2
**Presentation:** 1
**Contribution:** 1
**Rating:** 2
**Confidence:** 4

**Summary:**

This work reinterprets the previously proposed "feature selection layer", which was designed to give partially interpretable medical diagnoses, as a generic blackbox explainer approach which can be attached to the input of any model as a set of sparse linear weights.  Recovery of important features is demonstrated on simple synthetic datasets

**Strengths:**

Sparse linear weights as an easily applied method for quickly understanding important inputs is seemingly novel and is generally applicable to any model.

Results show that sparse selection demonstrates recovery of important features on small synthetic datasets

**Weaknesses:**

Although recovery results are established on easy synthetic datasets, a comparison of the benefits and disadvantages with respect to well-established explainability approaches is not explored in great detail.

**Questions:**

- What are the pros and cons of FSL compared with other existing explainability approaches?
- How long does FSL take to train in comparison with other posthoc XAI approaches?
- What is the purpose of the t-SNE plots and silhouette score?
- What is the intuition for what the silhouette score is measuring and why is it being used instead of more typical metrics like inclusion/removal curves or visualization of important features?

---

> ### Author Response · Authors · 2025-11-28
> **Answers to Reviwer SciF perceived weaknesses**
>
> We thank the reviewer for the constructive feedback and address the weaknesses points below.
>
> **Regarding the reliance on simple synthetic datasets.**
>
> Our experimental evaluation ranged from synthetic datasets, chosen because they allow for precise control over ground-truth informative features, to real-world high-dimensional, low-sample microarray datasets (line 605), such as the Breast dataset (151 samples and 54676 features).
> The proposed method achieved robust results in both scenarios, demonstrating particularly strong performance on the more complex real-world datasets.
>
> **Regarding the limited comparison with established explainability approaches and reliance on simple synthetic datasets.**
>
> To address the request for a comparison with well-established explainability approaches, we highlight the following trade-offs:
>
> Advantages of FSL:
>
> - Integrated Optimization: Unlike post-hoc methods, FSL can be utilized to improve the network's predictive accuracy simultaneously while learning the feature weights (**lines 320 and 374**).
> - High-Dimensional Performance: It demonstrates superior performance on complex, high-dimensional real-world datasets compared to standard baselines (**line 469**).
> - Simple implementation: Post hoc FSL is simple to implement and uses the same learning method of the model being interpreted (**line 114**).
>
>
> Limitations:
>
> - Model Specificity: The method is intrinsically designed for neural networks, whereas well-established approaches like SHAP or LIME are model-agnostic and can be applied to any classifier.
> - General Feature Weighting: Post hoc FSL is unable to capture class-specific feature weights (**line 476**).
> Designed for Tabular Datasets: Post hoc FSL is specifically designed for tabular data, where each feature has a fixed meaning within the dataset (**line 478**)

---

> ### Author Response · Authors · 2025-11-28
> **Answers to Reviwer SciF questions**
>
> We thank the reviewer for the insightful questions and address them below.
>
> **How long does FSL take to train in comparison with other posthoc XAI approaches?**
>
> Unfortunately, we did not include a formal execution time comparison, though we will address this in the paper. However, during experiments with high-dimensional datasets, the execution time of the compared methods was relatively higher, SHAP took 3 times more for datasets with huge amounts of features. Post hoc execution time increases with the complexity of the model.
>
> **What is the purpose of the t-SNE plots and silhouette score?**
>
> The primary objective of using Weighted t-SNE is to provide a global visual inspection tool to compare how different feature scoring techniques impact data clustering. Compared to other visualization techniques, t-SNE offers significant advantages that make it preferable for evaluating feature scoring, particularly its ability to capture non-linear relationships and its robust performance in revealing cluster separation.
>
> This method was originally proposed by (Grisci et al., 2025), and has been utilized in multiple studies as Gomes Junior and Lopes (2022) and Zhao et al. (2023).
>
> We will revise the paper to better explain the use of weighted t-SNE and the Silhouette score.
>
> **What is the intuition for what the silhouette score is measuring and why is it being used instead of more typical metrics like inclusion/removal curves or visualization of important features?**
>
> The intuition of silhouette score is to quantify the quality of cluster separation, the Weighted t-SNE results are presented alongside the Silhouette coefficient. This metric ranges from -1 to 1. A value close to 1 indicates that a point is well-matched to its assigned cluster, while a value of 0 implies that the point is near a decision boundary, and -1 suggests that the point may have been assigned to the wrong cluster.
>
> We acknowledge that additional experiments, such as feature erasure (where features identified as most relevant are removed to measure the impact on model performance), would provide complementary insights. However, this does not alter the fact that the compared methods and the proposed model are capable of extracting distinct levels of information from the original model. Notably, in the majority of cases, Post-Hoc FSL identifies features that result in superior class separation, despite the other methods utilizing the identical base model and data.

---

### Official Review · Reviewer_DYcX · 2025-10-31

**Soundness:** 1
**Presentation:** 1
**Contribution:** 1
**Rating:** 2
**Confidence:** 5

**Summary:**

The paper aims at introducing an updated post-hoc version of Feature Selection Layer. The proposed algorithm inserts a series of independent weights at the input data and learns these weights by freezing the pre-trained model with L1 regularization loss. These weights are then the representations of the importance of a feature in a specific dataset. The model is tested on a series of tabular classification datasets and across different metrics.

**Strengths:**

The method provided is fairly simple and according to the authors results seems to work in a simple environment.

**Weaknesses:**

1. The algorithm, as highlighted by the author only works in simple, tabular binary classification problems and fails in multi-class where it is unable to identify local and class specific feature’s relationships.
2. The algorithm produces some dataset level feature explanations (this is due by the fact that the weights are trained once and remain constants for all the samples). While most of the compared algorithms produce sample wise feature explanations. These two might be similar only in fairly consistent dataset and explains why the algorithm doesn’t work on more complex dataset or non tabular data.
3. The paper overlooks a lot of details, some of these make the paper irreproducible and leaving the reader with a lot of doubts (the author states that the code will be made available if accepted, nonetheless a lot of crucial details are missing to better understand and reproduce the paper results from the paper):
    * The dataset splits are not highlighted and it is not clear on what the model is finetuned.
    * The architecture of the pretrained model is not specified. Although at line 355 the authors talks about a “complement” to their experiments by using a pretrained model (TabPFN), this doesn’t seem to be the architecture they used on the previous results.
    * The model seems to be tested on a single trained model and a single pretrained one (this is not enough for post-hoc methods)
    * It seems like the pretrained model is trained by the authors but it is not clear what hyperparameter they used to do so or what kind of split, optimizer, number of epochs, etc.
    * Similar applies to the finetuning of the weights.
    * Training and finetuning curves and graphs are missing
4. A few minor errors:
    * Multiple time the author talks about the FSL (both post-hoc and original) as a dense layer, this is not the case as they are simple elements-wise weights for the input or it can be seen as a diagonal matrix (this misconception is present both in the related works at lines 83-84 and in the proposed implementation line 121-122)
    * Equation 2 at lines 164 applies the activation function only to the weight and not at the element-wise multiplication of the weights and the input features x.
    * Figure 1 could be represented following the standard neural network representations with weights on the edges, resulting in a clearer description.

**Questions:**

These are more curiosity or possible reflection points for improvements.
1. What about using different architectures (both pretrained and to train)?
2. Have you tried to normalize the learnable weights so that they sum to one?
3. why not directly using the first layer of the pretrained network and combining the weights for each input?
4. Why not for each sample freezing everything beside the first layer and temporarily finetune on a single sample and then merging the weights? (Going something along this line of direction would resolve the problem of computing dataset vs samples feature explanation)

---

> ### Author Response · Authors · 2025-11-28
> **Answers to Reviwer DYcX perceived weaknesses**
>
> We thank the reviewer for the constructive feedback and address the weaknesses points below.
>
> **Regarding the limitation to identify local and specific feature’s relationships and working only in tabular binary classification problems**
>
> The limitation to identify local and specific feature’s relationships was explicitly acknowledged in the conclusions regarding future work (**line 476**). However, as shown in Figure 7 (**line 772**) and Table 6 (**line 432**), for the Breast dataset (which contains 6 classes), the proposed method was capable of detecting the most important features. The Post-Hoc Feature Selection Layer exhibits significantly superior performance in generating meaningful feature attributions, having better results in cluster separation when compared to the other post-hoc methods.
>
> Furthermore, our method is applicable to regression problems and is not limited strictly to binary classification tasks.
>
> **Regarding the method’s limitation in tabular datasets**
>
> Our method handles complex tabular datasets, including high-dimensional ones with over 50,000 features. The restriction to tabular data arises from the method's nature as a global feature attribution approach, which requires features to possess a fixed meaning across all samples.
>
> While local methods give interpretable explanations for individual predictions, our method aims to learn a set of weights based on how each feature is important across all samples.
>
> **Regarding the overlook of details**
>
> Unfortunately, the code cannot be made available earlier because it is a way of identifying the authors. The splits were not mentioned in the paper. The specific test split percentages used for each dataset were:
>
> | DATASET | TEST SPLIT |
> | :--- | :---: |
> | XOR | 20% |
> | Synth | 10% |
> | Spam | 20% |
> | Liver | 10% |
> | Breast | 10% |
>
> We trained 10 MLP models (2 per dataset: one with the original embedded FSL and one standard MLP to be interpreted by post-hoc methods) and one external model, TabPFN (Transformer-based). Our method was evaluated across different architectures with varying depths and parameters.
>
> Our models were finetuned using the same training dataset and were trained using the Adam optimizer with a learning rate of 0.001 and weighted Cross Entropy loss. The specific batch sizes and epochs per dataset were:
>
> | DATASET | EPOCHS | BATCH SIZE |
> | :--- | :---: | :---: |
> | XOR | 50 | 32 |
> | Synth | 45 | 16 |
> | Spam | 80 | 16 |
> | Liver | 250 | 8 |
> | Breast | 250 | 8 |
>
> The missing details will be added in the final paper.
>
> Regarding training curves, these were initially omitted to prioritize final performance metrics within the page limit, however they can be added in the final paper.
>
> **Regarding the few minor errors**
>
> We acknowledge that the definition provided in Section 2.2 (**line 083**) regarding the introduction of the FSL Layer was imprecise. The FSL is a simple layer where each input feature connects to exactly one corresponding node (a one-to-one connection).
>
> Regarding the activation function, it is applied exclusively to the weights, with the specific objective of eliminating negative values (as detailed in Section 3.3 **line 171**). We also agree with the reviewer that Figure 1 can be improved as suggested.

---

> ### Author Response · Authors · 2025-11-28
> **Answers to Reviwer DYcX questions**
>
> We thank the reviewer for the **insightful questions** and the **interesting proposed ideas**.
>
> **What about using different architectures (both pretrained and to train)?**
>
> We used 5 different MLP architectures and a transformer-based architecture TabPFN to evaluate Post hoc FSL. In future experiments we plan to use more different architectures beyond the MLPs.
>
> **Have you tried to normalize the learnable weights so that they sum to one?**
>
> We started with this assumption, however for complex datasets with more features, as Breast that has more than 50.000 features, this resulted in too low weights per feature, affecting the training of the models negatively. The original FSL proposal had a weight  initialization of 1/n, where n is the number of features. Given this original initialization the model was incapable of learning on complex datasets as Breast due to the small weights (1/50.000). Therefore, we adopted our current initialization strategy to ensure training stability in high-dimensional settings.
>
> **Why not directly using the first layer of the pretrained network and combining the weights for each input?**
>
> That is an interesting idea for future experiments. We could introduce a new regularization technique based on this approach; however, it would require us to re-train the entire network.
>
> **Why not for each sample freezing everything beside the first layer and temporarily finetune on a single sample and then merging the weights? (Going something along this line of direction would resolve the problem of computing dataset vs samples feature explanation)**
>
> This is an interesting suggestion for bridging the gap between dataset level and sample level explanations. Investigating whether the Post hoc FSL can be adapted for a single sample feature weighting learning via fine-tuning is a promising direction for future work. While this specific experiment falls outside the current scope, we intend to explore this approach in future work to access its feasibility.

---

### Official Review · Reviewer_qq6z · 2025-10-31

**Soundness:** 1
**Presentation:** 1
**Contribution:** 1
**Rating:** 2
**Confidence:** 4

**Summary:**

This paper proposes an adaptation of the Feature Selection Layer (training process) to the context of post-hoc feature attribution for tabular data. The proposed method consists of freezing the neural network to be explained and placing a linear layer between the input and the first layer of the frozen architecture, where each neuron has a 1-to-1 correspondence with an input feature. The augmented network is trained to replicate the original network’s outputs while keeping the original network frozen (i.e., only the added layer is trained). The idea is that, at the end of training, the weights between the input and this layer can be used to express the importance of each individual feature. The method is evaluated using metrics related to pure performance (e.g., accuracy and F1) against the original model and the FSL method and, in terms of stability and separability against FSL and post hoc feature attribution methods, finding that the proposed method is in general able to identify relevant features outperforming competitors in separability and visual metrics.

**Strengths:**

- While the addition of a layer at the end of a pre-trained model is common in explainability literature, adding a layer at the beginning of the network to achieve the same goal is interesting.

**Weaknesses:**

In general, the paper suffers from a duality between feature selection and global feature attribution. The paper states that the goal of the proposed method is to “highlight the features the original model considers most important,” which falls under global feature attribution and interpretability. This differs from feature selection, which would require a different setup and comparison. The following review is based on this assumption, and the method is reviewed as a post hoc global feature attribution approach.

- There is a **mismatch between the claims and what is currently demonstrated** in the paper. **This method is not post hoc**. Post hoc methods do not modify the network’s decision process. While the method keeps the original model’s weights frozen, the network’s response (i.e., the activations of the frozen part) will differ from the original model. This can be verified by comparing the activations of the frozen network before and after training the added layer. If they differ, even marginally, then the decision process is different. In this context, the authors’ statement that the layer *“enhances interpretability and potentially improves predictive performance”* implicitly confirms that the method cannot be considered post hoc, since performance can change.

- **Weak evaluation setup: almost all metrics applied are better suited to feature selection rather than feature attribution**. For some of them, it is not clear how they are applied to feature attribution methods (see below). Feature attribution is a major area in explainability, and there are many metrics and procedures for global feature attribution methods. For example, visual separability metrics (employed by the authors) are appropriate for feature selection, since the goal is to select informative features that capture most of the state space. Conversely, in feature attribution, separability is not a goal, since the model could consider just a couple of highly correlated features as important due to shortcut learning. There are also concerns related to “ground-truth rankings”. While effective for feature selection, in explainability contexts the model may learn different rankings (which could explain the stability issues observed by the authors), highlighting that these two fields require different evaluation setups. In this context, the fact that the proposed method outperforms feature attribution methods only on separability metrics is an additional factor that limits the paper’s significance and contribution.
- **Limited novelty**: as stated by the authors, the paper does not modify the original FSL but proposes a slightly different training paradigm. The difference, from my understanding, lies in the initialization of the weights (to 1), the use of ReLU as the activation function, and keeping the main network frozen. In this context, the novelty seems very limited.
- **Missing details**: there are several missing details on how global feature attributions are computed for competitors (e.g., for stability). For example, are attributions computed as a sum over the full dataset or as an average? What is the value of K in K-fold (I could have missed it)?
- The **“Related Work” section is limited**. Only five methods (those used for comparison) are briefly cited, and the most recent among them is from five years ago. The paper should better contextualize the method within current and recent literature.


Minors: the post hoc methods used as comparisons are better characterized as baselines rather than state-of-the-art methods, given advances in feature attribution over the last five years.

**Questions:**

see weaknesses

---

> ### Author Response · Authors · 2025-11-28
> **Answers to Reviwer qq6z perceived weaknesses**
>
> We thank the reviewer for the constructive feedback and address the weaknesses points below.
>
> **Regarding the duality between feature selection and global feature attribution**
>
> We adopted the term “Post-hoc Feature Selection Layer” primarily to maintain consistency with the method we adapted (Figueroa Barraza et al., 2021), which designates it as a “Feature Selection Layer”. However, we agree with the reviewer that our method functions as a **global feature attribution** mechanism, and we will revise the paper to better clarify this distinction and ensure precise terminology throughout.
>
> Crucially, we emphasize that our experimental comparisons were conducted against other feature weighting methods; therefore, the change in terminology does not invalidate the experimental setup or the results.
>
> **Regarding the post hoc aspect of the method**
>
> We employed the terminology “**Post-Hoc**” specifically to distinguish our approach from the original Feature Selection Layer implementation, which requires joint training with the network. Our method is applied strictly after the model has been fully trained.
>
> Our method uses the pre-learned internal weights to highlight which features the network finds most important without altering these internal parameters. After the analysis, we obtain a list of weights for each feature that defines their importance, which is also capable of improving the network's performance by maximizing relevant features and minimizing noise. Furthermore, we can remove our method from the network, causing the pre-trained model to revert to its state before being integrated with the Post-Hoc FSL. We will update the paper to explicitly clarify these points and remove any ambiguity regarding the terminology.
>
> **Regarding the weak evaluation setup**
>
> The primary objective of using Weighted t-SNE is to provide a global visual inspection tool to compare how different feature scoring techniques impact data clustering (Grisci et al., 2025). Compared to other visualization techniques, t-SNE offers significant advantages that make it preferable for evaluating feature scoring, particularly its ability to capture non-linear relationships and its robust performance in revealing cluster separation.
>
> We acknowledge that additional experiments, such as feature erasure (where features identified as most relevant are removed to measure the impact on model performance), would provide complementary insights. However, this does not alter the fact that the compared methods and the proposed model are capable of extracting distinct levels of information from the original model. Notably, in the majority of cases, Post-Hoc FSL identifies features that result in superior class separation, despite the other methods utilizing the identical base model and data. Furthermore, it is important to highlight that the Post-Hoc FSL can be leveraged to explicitly improve the network's accuracy while simultaneously learning the importance weights of each feature.
>
> **Regarding the limited novelty**
>
> The novelty of our work lies in the adaptation of the original FSL method to a new context, rather than the creation of a completely new architecture from scratch. This paradigm shift significantly broadens the method's applicability, allowing FSL to be integrated into pre-existing models, as demonstrated by our pre trained models and the TabPFN experiments. This capability offers dual advantages: it enables the extraction of interpretability knowledge (feature relevance) and enhances the classification performance of the base network.
>
> Furthermore, the proposed improvements, such as the initialization of the FSL weights to 1, were crucial for the deployment of our method in high-dimensional, low-sample size datasets, such as microarray datasets.
>
> **Regarding missing details**
>
> Global feature attributions for the competing methods were calculated using the average. The values for the number of folds used can be found in **APPENDIX A, DATASET DESCRIPTION** (**line 605**). The missing details will be added to the original text.
>
> **Regarding the limited “Related Work” section**
>
> We prioritized methods that are well-established and consolidated within the field of neural network interpretability. We selected these specific baselines because they are proven to deliver high-quality interpretability results and also share the post-hoc nature of our proposed method, ensuring a fair and direct comparison.

---

### Meta-Review · Area_Chair_86L8 · 2025-12-10

**Summary:**

The authors propose an extension to the FSL layer for post-hoc feature attribution of tabular data. The reviewers acknowledge the simplicity of the proposed method but had several significant concerns, most notably:

- mismatch between the claims and what is currently demonstrated `[qq6z]`
- weak evaluation setup `[qq6z]`
- limited novelty `[qq6z]`
- missing details `[qq6z, DYcX]`
- weak results on more complex tasks, e.g., multi-class `[DYcX]`
- missing comparison with alternative explainability methods `[SciF]`

The authors addressed some concerns during the rebuttal, but some core aspects remain not sufficiently resolved (e.g., limited novelty, weak evaluation).

**Reviewer Concerns:**

During the rebuttal, the authors addressed the mentioned concerns. Some of them can be considered as (at least partially) resolved:
- mismatch between the claims and what is currently demonstrated `[qq6z]`
- missing details `[qq6z, DYcX]`

But several concerns are not sufficiently addressed, namely:
- weak evaluation setup `[qq6z]`
- limited novelty `[qq6z]`
- weak results on more complex tasks, e.g., multi-class `[DYcX]`
- missing comparison with alternative explainability methods `[SciF]`

**Reviewer Scores:**

Most probably, all reviewer scores have remained unchanged.

---

### Decision · Program_Chairs · 2026-01-26

Reject